# Viral RNA Metagenomics of *Hyalomma* Ticks Collected from Dromedary Camels in Makkah Province, Saudi Arabia

**DOI:** 10.3390/v13071396

**Published:** 2021-07-18

**Authors:** Fathiah Zakham, Aishah E. Albalawi, Abdullah D. Alanazi, Phuoc Truong Nguyen, Abdulaziz S. Alouffi, Altaf Alaoui, Tarja Sironen, Teemu Smura, Olli Vapalahti

**Affiliations:** 1Department of Virology, Faculty of Medicine, University of Helsinki, 00014 Helsinki, Finland; phuoc.truong@helsinki.fi (P.T.N.); tarja.sironen@helsinki.fi (T.S.); teemu.smura@helsinki.fi (T.S.); olli.vapalahti@helsinki.fi (O.V.); 2Department of Veterinary Biosciences, Faculty of Veterinary Medicine, University of Helsinki, 00014 Helsinki, Finland; 3Faculty of Pharmacy, University of Helsinki, 00014 Helsinki, Finland; 4Department of Biology, Faculty of Science, University of Tabuk, Tabuk 47912, Saudi Arabia; ae.albalawi@ut.edu.sa; 5Department of Biological Sciences, Faculty of Science and Humanities, Shaqra University, Ad Dawadimi 11911, Saudi Arabia; aalanazi@su.edu.sa; 6King Abdulaziz City for Science and Technology, Riyadh 12354, Saudi Arabia; aaloufi@kacst.edu.sa; 7Laboratory of Engineering Sciences and Modeling, Ibn Tofail University, Kenitra 14000, Morocco; altaf.alaoui@gmail.com; 8HUS Diagnostic Center, HUSLAB, Hospital District of Helsinki and Uusimaa, Helsinki 00260, Finland

**Keywords:** virome analysis, *Hyalomma dromedarii*, dromedary camels, province of Makkah, Saudi Arabia, arboviruses, tick-borne diseases

## Abstract

Arthropod-borne infections are a medical and economic threat to humans and livestock. Over the last three decades, several unprecedented viral outbreaks have been recorded in the Western part of the Arabian Peninsula. However, little is known about the circulation and diversity of arthropod-borne viruses in this region. To prepare for new outbreaks of vector-borne diseases, it is important to detect which viruses circulate in each vector population. In this study, we used a metagenomics approach to characterize the RNA virome of ticks infesting dromedary camels (*Camelus dromedaries*) in Makkah province, Saudi Arabia. Two hundred ticks of species *Hyalomma dromedarii* (*n* = 196) and *Hyalomma impeltatum* (*n* = 4) were collected from the Alkhurma district in Jeddah and Al-Taif city. Virome analysis showed the presence of several tick-specific viruses and tick-borne viruses associated with severe illness in humans. Some were identified for the first time in the Arabian Peninsula. The human disease-associated viruses detected included Crimean Congo Hemorrhagic fever virus and Tamdy virus (family Nairoviridae), Guertu virus (family Phenuiviridae), and a novel coltivirus that shares similarities with Tarumizu virus, Tai forest reovirus and Kundal virus (family Reoviridae). Furthermore, Alkhurma hemorrhagic virus (Flaviviridae) was detected in two tick pools by specific qPCR. In addition, tick-specific viruses in families Phenuiviridae (phleboviruses), Iflaviridae, Chuviridae, Totiviridae and Flaviviridae (Pestivirus) were detected. The presence of human pathogenetic viruses warrants further efforts in tick surveillance, xenosurveillence, vector control, and sero-epidemiological investigations in human and animal populations to predict, contain and mitigate future outbreaks in the region.

## 1. Introduction

Recently, Saudi Arabia has witnessed an upsurge of viruses with high morbidity and mortality rates in humans and livestock. Some of these, e.g., the Middle East Respiratory Syndrome (MERS) virus and Alkhurma hemorrhagic fever virus (AHFV), emerged for the first time, while others, e.g., dengue virus, Crimean-Congo hemorrhagic fever virus (CCHFV), Kadam virus (KADV), West Nile virus, Rift Valley fever virus and chikungunya virus, have re-emerged after spreading in other parts of the world [1,2,3].

Most emerging human viral diseases are of zoonotic origin, and 40% of these are vector-dependent [4]. Ticks are hematophagous vectors capable of transmitting viruses to human and animal hosts. Several studies have found pathogenic viruses from different species of ticks in Saudi Arabia. For instance, CCHFV was isolated from *Hyalomma, Rhipicephalus* and *Amblyomma* ticks infesting small ruminants imported from Sudan to Saudi Arabia in 2000 [5]. In 2007, AHFV RNA was detected in *Ornithodoros savignyi* ticks from the Western part of Saudi Arabia after several human cases were reported [6]. Another study confirmed the circulation of the AHFV in *O. savignyi* and *H. dromedarii* ticks in the Najran region (southwestern part) of the kingdom [7]. Sindbis virus (SINV) was isolated from *Hyalomma* species (*H. dromedarii* and *H. impeltatum*) in the Al-Qassim, Riyadh and Jazan regions [8]. Dhori virus was also isolated and identified (*H. impeltatum* and *H. schulzei*) in the eastern region [8]. In the early 1980s, KADV (*Flavivirus*) was detected in a pool of male *H. dromedarii* ticks taken from a dead camel in Wadi Thamamah in Riyadh [3]. Before that, KADV was isolated from *Rhipicephalus* and *Amblyomma* ticks in Uganda and Kenya and was serologically associated with human and cattle infection. Another study confirmed the isolation of KADV from *H. dromedarii* and *H. anatolicum* in the Al-Qassim region of Saudi Arabia [8].

*Hyalomma* ticks are common in the Arabian Peninsula and other Asian countries [9,10]. They belong to the family of hard ticks *Ixodidae* and ectoparasitize camels. Dromedary or Arabian camels are the main domesticated animals in Saudi Arabia and are of economic and medical importance. Recently, camels have been associated with the transmission of several viral diseases to humans. KADV, AHFV, CCHFV, Tamdy virus (TOV) and MERS are associated with camels [3,11,12,13]. For instance, Hemida et al. found that 90% of camel populations were MERS-CoV seropositive [13]. Recently, a novel lineage of CCHFV has been identified in the nasopharyngeal samples collected from camels in the United Arab Emirates [11]. Dromedary camels are also associated with prions and severe forms of neurodegenerative prion disease [14] as well as several bacterial pathogens [9].

In this study, we identified viral pathogens from ticks infesting dromedary camels in the Makkah province of Saudi Arabia using a metagenomic approach [15]. Earlier studies have shown that metagenomic approaches can be used to detect both known and novel viruses associated with animal and human diseases [16,17]. Such approaches can enable timely responses to vector-borne disease outbreaks and help in preparedness as well as the implementation of practical strategies and policies to contain and mitigate the threat of these infections.

## 2. Research Methodology

### 2.1. Makkah Region

The Makkah (Mecca) region is located on the western Arabian Peninsula. It includes important cities such as Makkah (Mecca), the holiest city in the Islamic world, Jeddah, the biggest sea port on the Red Sea and Al-Taif, with a population of 8.8 million (Figure 1). Millions of Muslim pilgrims gather and slaughter more than 1.2 million sheep, goats and camels annually during their pilgrimage [18], as a part of their religious rituals. This may enhance the transmission of infectious diseases, including zoonotic and vector-borne diseases. The climate of Makkah is arid, and most animals that live in the region are adapted to this aridity.

### 2.2. Tick Collection

From the mid-July to October 2019, a total of 200 camels were screened for ticks in the western part of Saudi Arabia, Al-Taif (Coordinates: 21°16′30.34″ N 40°24′22.16″ E) and Al-Khurma (21°55′ N 42°02′ E) (Figure 1). Each camel was apparently healthy at the time of sampling. Ticks found within 15 min were collected (1 tick per infested animal), and a total of 200 ticks (100 ticks from each locality) were collected and placed in separate tubes, containing RNAlater. The samples were shipped to the University of Helsinki, Finland, for further analysis.

### 2.3. Tick Species Identification

Ticks were morphologically identified to genus level. Two legs from each tick were separated for DNA extraction and barcode sequencing. DNA was extracted using GeneJET Genomic DNA purification kit (Thermo Scientific). A 604 bp fragment of the cytochrome c oxidase subunit 1 (cox1) gene was amplified as described previously [19,20]. The Phusion Flash High-Fidelity PCR Master Mix was used for amplification, and the thermal cycler was programmed as follows: 10 s at 98 °C, 30 cycles of 1 s at 98 °C, 5 s at 55 °C, 15 s at 72 °C and a final extension step of 1 min at 72 °C. The PCR products were purified using the GeneJET PCR Purification Kit (Thermo Fisher, Waltham, MA, USA) and submitted for Sanger sequencing at the Finnish Institute for Molecular Medicine (FIMM), University of Helsinki. The sequences obtained were compared to other sequences deposited in GenBank using the NCBI Basic Local Alignment and Research Tool (BLAST) program. The cox1 sequences were deposited at Genbank under the accession numbers (MZ348624-MZ348823).

### 2.4. RNA Extraction and Pooling

Each tick was mechanically homogenized with Tissuelyser for 30 cycles/s with 1 mL of TRIzol LS reagent and two 5 mm stainless steel beads. The total RNA was extracted from the homogenate according to the manufacturer’s instructions. RNAs were combined into 20 pools (10 samples per pool). Ten pools contained ticks collected from Al-Khurma (pools K1 to K10) and another ten pools contained ticks collected from Al-Taif (pools T1 to T10). The RNAs from engorged and non-engorged ticks were divided into separate pools. The three pools from each locality contained engorged ticks (K1 to K3 and T1 to T3), and the rest of pools contained non-engorged ticks.

### 2.5. Pathogen-Specific PCRs

To confirm the presence of viral pathogens, a previously described PCR method for *Phlebovirus* detection [21] and qPCR method for AHFV detection [22] were used. After sequencing, we found some TOV contigs and developed a real time PCR method targeting the L segment of TOV using the forward primer 5′ACACGTTTCTTGGGAGATGC3′, reverse primer 5′GAGCTTGCGCTGCTTTTATT3′ and the probe FAM-CAAGGACCATGAGACTGCTG-BHQ. TaqMan Fast Virus 1-Step Master Mix (Thermo Scientific, Waltham, MA, USA) was used for amplification. The positive samples for TOV were also checked by a nested PCR for Tacheng tick virus 1 (TcTV1), as previously described [23].

### 2.6. Library Preparation and Sequencing

Ribosomal RNA was removed using the NEBNext rRNA Depletion Kit, and the sequencing libraries were prepared using NEBNext Ultra II RNA Library Prep Kit (New England BioLabs) according to the manufacturer’s protocol. The library concentrations were measured by Qubit dsDNA BR Assay Kit (Life Technologies, Carlsbad, CA, USA), and the fragments sizes were checked by gel electrophoresis. Sequencing was carried out using MiSeq Reagent Kit V2 with 150 bp reads.

### 2.7. Sequence Analysis

The raw sequence reads were quality-filtered, de novo assembled and annotated using Trimmomatic [24], Megahit [25] and SANSparallel [26] programs, respectively, implemented in Lazypipe pipeline [16]. The unidentified contigs from Lazypipe were further sought for virus matches by Blastx followed by open reading frame search using EMBOSS getorf [27]. For the phylogenetic analysis, complete amino acid sequences of all related viruses were retrieved from GenBank (accessed in May 2021). Multiple sequence alignment of amino acid sequences was performed using ClustalW algorithm followed by manual trimming. Protein similarity matrix was done for the S segments (the segment with highest coverage) of the known human pathogenic virus species using Sequence Demarcation Tool (SDT) v1.2 [28]. Phylogenetic analysis was performed using the maximum likelihood method implemented in IQ-TREE2 [29] using ModelFinder [30] and ultrafast bootstrapping with 1000 replicates [31].

## 3. Results

### 3.1. Tick Species Identification

Morphologically, all the ticks belonged to *Hyalomma* spp. A commonly used barcode gene cox1 was amplified and partially sequenced. BLASTn search of the sequences showed that the ticks in this collection were camel ticks, *H. dromedarii* (*n* = 196, 98%) and *H. impeltatum* (*n* = 4, 2%), with 98 to 100% similarity with sequences available at GenBank.

### 3.2. Virome Analysis

Most of the sequence reads (3,258,467; 90.2%) were from eukaryotes, followed by viruses (177,167; 5%), bacteria (175,262; 4.8%) and bacteriophages (465). The RNA virome analysis (Table 1) showed the presence of several virus families, of which the most commonly detected virus genus was *Phlebovirus*, family *Phenuiviridae* (1.8% of the total reads).

### 3.3. Nairoviridae

Members of genus *Orthonairovirus* were found in pools K1, K4 and K10 from the Alkhurma district. The K1 pool showed sequences similar to CCHFV, with identities of 99.67%, 98.64 % and 96.88% to S, M and L segments, respectively, to strain camel/Abu Dhabi/B84, isolated from the nasopharyngeal sample of a camel in Abu Dhabi, United Arab Emirates. In these, Partial S (residues 184–482), M (residues 287–560 and 1214–1507) and L segments (different contigs of which the longest fragment covered residues 958–1307) were found. Consistently, the protein matrix of different nairoviruses S segments and the phylogenetic analysis of the S,M,L segments of the detected sequences grouped together with CCHFV sequences from United Arab Emirates, Sudan and South Africa (Figure 2 and Figure 3).

Another *Orthonairovirus*, Tamdy virus (TOV), was detected in pools K4 and K10 from Alkhurma District. Fragmental S (residues 278–483), M (142–304) and L (different fragments of which the longest covered residues 1526–1887) segments were found in the pool K4, and fragmental L segment (residues 1619–1805) was found in pool 10. Real-time PCR also identified pool K9 positive, and when performed on individual ticks of TOV-positive pools, showed that 10% (10/100) of the ticks collected in the Alkhurma region were positive for TOV. The sequences showed 95–99% identity to TOV strains detected in *H. asiaticum* ticks infesting sheep in Armenia, Azerbaijan and Uzbekistan as well as the recently identified strain TOV-XJ01 from China that has been found in ticks collected on Bactrian camels. In addition, we screened the samples for newly identified pathogen TcTV1 using nested PCR, but found no TcTV-1 positive ticks. The phylogenetic trees of S, M and L segments are shown in Figure 3.

### 3.4. Phenuiviridae

Most *Phenuiviridae* found in this study were apparently tick-specific phleboviruses, such as Iftin virus (IFTV). However, the T3 pool from Taif showed sequences similar to *Banyangvirus* (Guertu virus, GTV, and Severe fever with thrombocytopenia virus, SFTV) isolated in China.

#### 3.4.1. Banyangvirus

The contig lengths of *Banyangvirus* sequences ranged between 465 and 1737 and mapped and clustered segments L, S and M of GTV. The nucleocapsid sequence of the S segment was complete (245 aa) and shared 93% similarity with GTV strain DXM (YP_009666940.1) (Figure 4, Appendix A). The M segment (residues 232–1070) included partial Gn and most of Gc coding sequences and showed a similarity of 87% with GTV. The L segment (residues 673–1316) shared a similarity of 95% with GTV-DXM. Phylogeny analysis confirmed this finding (Figure 5).

#### 3.4.2. Phlebovirus

Most of the virus sequences detected belonged to the genus *Phlebovirus*, and most of the contigs represented the L segment (Appendix A). The identified sequences had the highest similarity to IFTV L segment (90.68–100%), which has been identified in *H. dromedarii* ticks collected from camels in Kenya (QSR83608.1) (Appendix A). A nearly complete L segment sequence was found in pool 3 (residues 31–2148 aa). In addition, several partial L segment sequences of IFTV were found in other pools from Alkhurma and Al-Taif regions (K1-K10, T1, T2, T5, T6, T7).

### 3.5. Reoviridae

Sequences mapping to genus *Coltivirus* segments VP1 to VP6, VP8 and VP9 were found in three pools K1-K3 from the Alkhurma region. These showed 39.68–65.07% amino acid similarity with Tarumizu virus (TARV), 43.57–71.37% similarity with Kundal virus (KUNDV), and 30.84–70.92% similarity with Taï Forest reovirus (TFRV) identified in ticks from Japan and India, and African tailed bats in Cote d’Ivoire, respectively. The high amino acid divergence suggests that the detected contigs belong to a novel coltivirus species. The highest similarity with TARV, KUNDV and TFRV was found with the VP1 gene encoding the RNA-dependent RNA polymerase (Figure 6). All the three pools included the segments VP1 to VP4. The VP5 segment was found in pools K1 and K2. A partial sequence of VP6 gene encoding the nucleotide binding NTPase was found only in pool K1. VP8 was found in two pools (K2 and K3) and VP9 was detected only in pool K3. Most of the segments were partial, except for the segment VP3 in pool K3 (residues 1–1164 aa) and VP4 in the same pool, which was nearly complete (19–1038 aa).

In the phylogenetic analysis, the detected coltivirus clustered together with KUNDV, TARV and TFRV in all detected segments, except for VP6, where TFRV sequence was not available. This group clusters together with Colorado tick fever virus (CTFV) and Eyach virus (EYAV). Furthermore, the novel coltivirus formed a sister group to KUNDV and TARV based on all other segments, except for VP4, where it was most closely related to TFRV, and segments VP2 and VP9, where the clustering remained unresolved (Figure 6).

### 3.6. Flaviviridae

AHFV was detected in two pools in the Alkhurma district and Al-Taif city (K2 and T9) by qPCR, with very low copy numbers (Ct value of 42), and no AHFV sequences were identified by NGS. However, some sequences belonging to *Pestivirus* were found in pool K3 from the Alkhurma district. The sequences share a similarity of 81.10% with a Bole Tick virus 4 from China and 79.55% with Trinbago virus (TBOV) found in ticks from Trinidad and Tobago [32] (Appendix A).

### 3.7. Chuviridae

Sequences belonging to the family *Chuviridae* were detected in one pool (K1) from the Alkhurma district. With a coverage of 80%, the sequences share high similarity with the RNA polymerase sequences of the unclassified Liman tick virus (95.7%) and *Mivirus*, Bole tick virus 3, with 74.8% identity (Appendix A).

### 3.8. Totiviridae

Two pools from the Alkhurma district (K1 and K3) showed sequences belonging to the family *Totiviridae*. With coverages of 99% and 92%, both sequences have identities of 42.65% and 55.56%, respectively, with the RNA polymerase sequences of Lonestar tick totivirus (Appendix A).

### 3.9. Iflaviridae

Sequence reads mapping to family *Iflaviridae* (1.4% of the total reads) were the second most common virus group after phleboviruses. Sequences longer than 1000 aa detected in pools K1 to K4 showed a high similarity with newly identified Iflavirus HdromIV (97–100% identity) [33]. Lower similarities (>75%) were found with other iflaviruses, like the Gerbovich virus and Hubei tick virus 1 and 2 (Appendix A).

## 4. Discussion

In the last few years, several outbreaks of zoonotic viruses have emerged in the Arabian Peninsula, with some being associated with high fatality rate in humans and/or animals. Ticks infesting vertebrates are a rich source of viruses, as demonstrated here in our investigation of the RNA virome of ticks from camels. Camels are important livestock in the region and have been associated with zoonotic diseases. While it is often not known whether camels are the main reservoir of pathogens or a spillover host, their role in the transmission of human pathogens should be studied. Furthermore, as little is known about the diversity of viruses in the ticks of the Arabian Peninsula, deepening the knowledge of tick-borne viruses is crucial to prepare for and contain future outbreaks or epizootics.

The ticks in this study were collected from the areas of Saudi Arabia where outbreaks of CCHFV and AHFV have occurred. The ticks were detached from dromedary camels and were identified as *H. dromedarii* and *H. impeltatum*, as in previous tick prevalence studies in Saudi Arabia [9]. Both species have been shown to harbor viruses such as CCHFV, ALKV, KADV and SINV [3,5,7,8]. We identified several known pathogens, such as CCHFV and AHFV, as well as potential pathogens related to recently described disease-associated viruses, such as TOV and GTV and a potential novel coltivirus. In addition, we detected several viruses that are most likely tick-specific. Notably, no coronaviruses such as MERS virus were detected.

Between 1989 and 1990, an outbreak of CCHFV occurred in the western province of Saudi Arabia. The virus was introduced via imported infected livestock [34]. Another report confirmed the isolation of CCHFV on cell culture from ticks collected from imported animals and showed fatal effects on experimental animals [5]. The virus detected in this study showed high similarity (96.88–99.67 to the CCHFV strains identified from samples collected from camels, suggesting that camels could be a potential reservoir of CCHFV [11]. In this study, yet another nairovirus, TOV, was identified for the first time in the Arabian Peninsula. TOV was first found in *H. asiaticum* and *H. plumbeum* collected between 1971 and 1974 from deserts of Uzbekistan and Turkmenistan [35]. It has also been detected in other central Asian countries, such as Azerbaijan, Armenia and Kazakhstan [36]. Recently, the virus was identified in *H. asiaticum* ticks infesting Bactrian camels in China [12] and in *H. marginatum* and *H. aegyptium* from Turkey [37,38]. The sequences identified in this study showed high similarity with the strains isolated in Eurasian countries and China, suggesting that TOV is geographically widespread. TOV has been shown to cause febrile illness in humans [39]. In addition, two virus species phylogenetically related to TOV, Tacheng tick virus 1 (TcTV1) and Songling virus, have caused febrile illness in humans [23,40]. Most recently, a novel tick-associated virus related to TOV, Sulina virus, was detected in *Ixodes ricinus* ticks in the Danube Delta, Europe [41].

Furthermore, a bandavirus that shares a high similarity (87–95%) with other pathogens of the genus *Banyangvirus* (GTV, SFTV and Heartland virus, HRTV) was detected in a tick pool from Al-Taif. This genus contains virus species that are highly pathogenic and have zoonotic potential [42]. For instance, SFTV and HRTV, found in Far Eastern Asia and the US, respectively, have been associated with severe fever and low platelet counts (thrompocytopenia), which may lead to death in some cases [43,44]. GTV is a novel pathogenic banyangvirus that was isolated from ticks in China [42]. It’s genome has 90% similarity to SFTV at the nucleotide level, and it is able to cause infection in experimental animals and human cell lines. A serological survey suggested that some people in Guertu County, China, have neutralizing antibodies against the virus [42]. This report demonstrates circulation of a banyangvirus in the Middle East region.

We also identified sequences that were related to potential pathogens in the *Reoviridae* family, including TARV, KUNDV and TFRV. The amino acid similarities ranged between 30.84 and 71.37%, which suggests that these sequences belong to a novel coltivirus species provisionally named “Jeddah tick coltivirus”. TARV was discovered in Japan in 2017, and the related KUNDV was identified recently in ticks from India. These group together with the pathogenetic CTFV and EYAV [45,46]. Although both TARV and KUNDV replicate in human cell lines, no human or animal cases due to these viruses have yet been detected [46]. Another member of this group, TFRV, was isolated from bats in Côte d’Ivoire. This virus shares a common ancestor with tick-borne coltiviruses, including the strains that we found in the current study [47].

In 1995, AHFV was first isolated in Saudi Arabia from six young men from the Alkhurma district [48]. Other reports have confirmed outbreaks of AHFV in different cities of the kingdom [49]. The circulation of AHFV was also confirmed in *H. dromedarii* ticks collected from the region of Najran [7]. Here, we report the presence of AHFV by qPCR in two pools, one from Al-Taif city and another from the Alkhurma district, but the results could not be confirmed with NGS, most likely due to the low copy number of the virus genome. Despite the robustness of NGS for the identification of a plethora of pathogens, it is less sensitive than qPCR for the detection of viruses [50]. Of the family *Flaviviridae*, pestivirus sequences, similar to Bole tick virus 4 and TBOV, were identified recently in ticks from in Trinidad and Tobago [32].

*Hyalomma dromedary* ticks harbored a plethora of virus groups, with a predominance of phleboviruses. Almost all the pools that showed viral sequences included the IFTV that has been found previously in *H. dromedarii* ticks in Kenya. Iflaviruses (*Iflaviridae*) are the second most prevalent group of tick-associated viruses in this collection of ticks. Other tick-associated viruses were also reported in this study, such as *Chuviridae* and *Totiviridae*. Totiviruses are among the endogenous viruses that are able to modulate arboviral infections in ticks and insects [51].

In conclusion, the virome of ticks collected from Saudi Arabia showed a number of pathogenic or potentially pathogenic virus groups. This emphasizes the need for tick surveillance in the region. Based on our findings, seroepidemiological investigations in the human population are warranted to assess the risk of zoonotic outbreaks in the region.

## Figures and Tables

**Figure 1 viruses-13-01396-f001:**
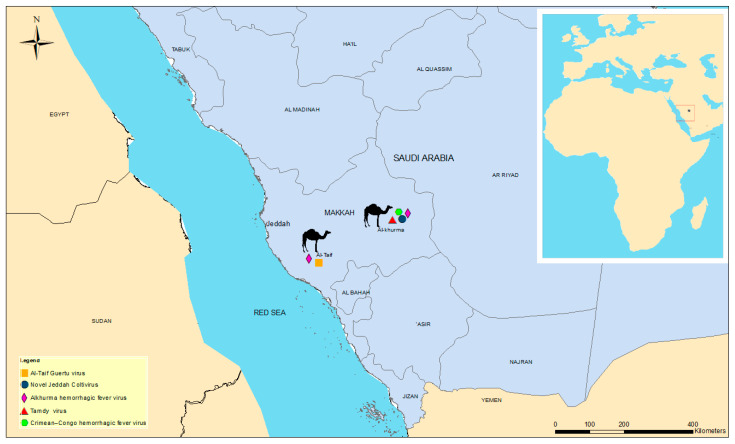
A map of the western part of Saudi Arabia showing the main identified potential pathogens in the province of Makkah.

**Figure 2 viruses-13-01396-f002:**
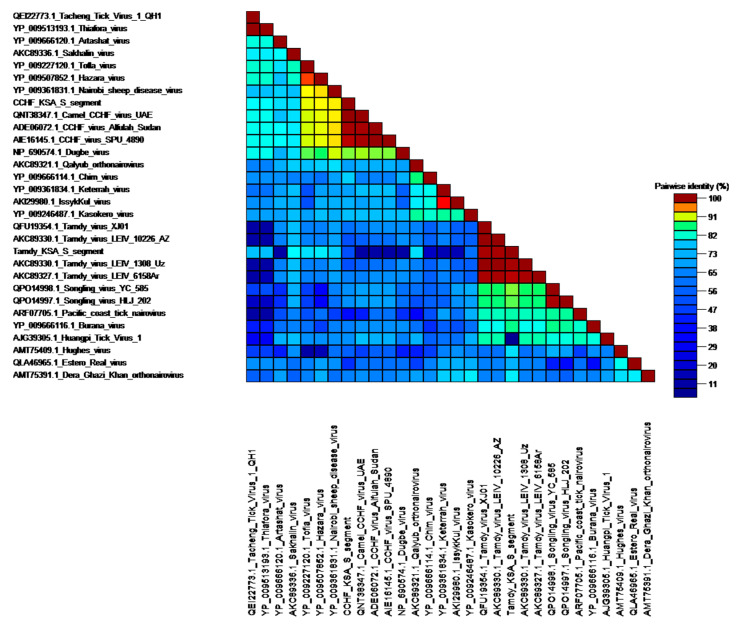
Pairwise protein identity matrix of nairoviruses including new strains identified in Saudi Arabia.

**Figure 3 viruses-13-01396-f003:**
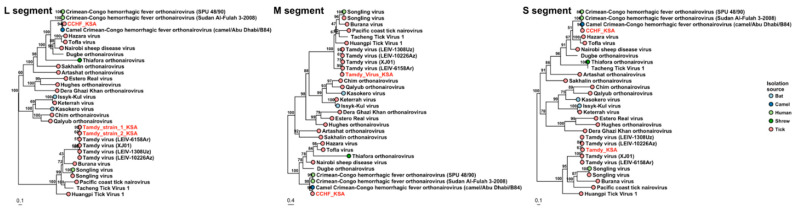
Phylogenetic trees based on L, M and S segments of nairoviruses. The viruses detected in Saudi Arabia are marked with red color.

**Figure 4 viruses-13-01396-f004:**
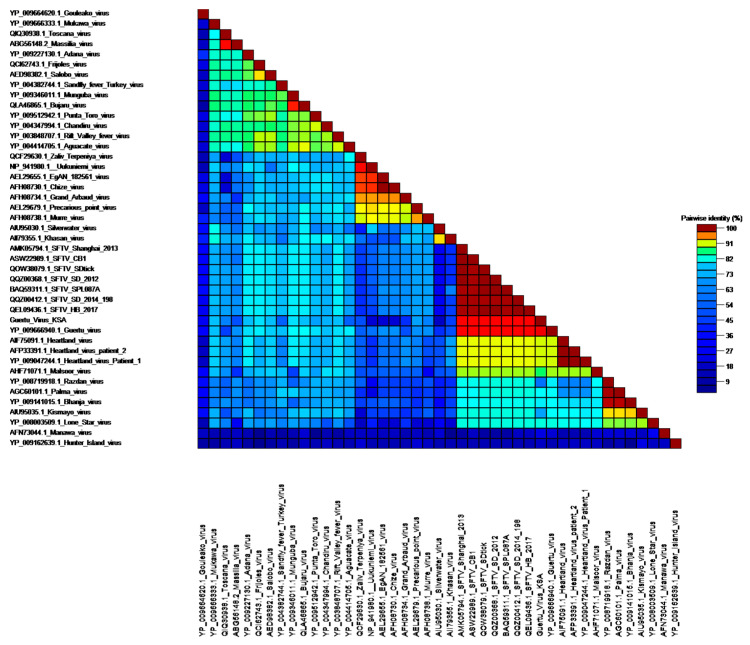
Pairwise protein identity matrix of *Phenuiviridae* S segment including new Guertu strain identified in Saudi Arabia.

**Figure 5 viruses-13-01396-f005:**
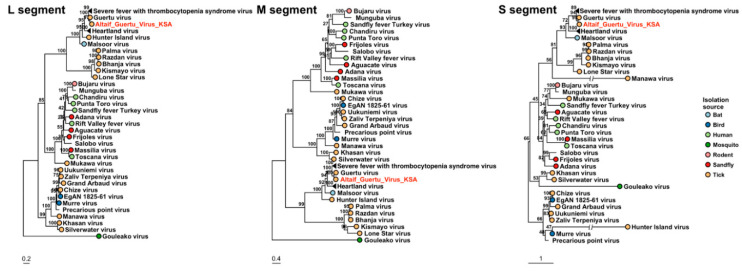
The phylogenetic trees based on L, M and S segments of the Bandaviruses (family *Phenuiviridae*). The viruses detected in Saudi Arabia are marked with red color.

**Figure 6 viruses-13-01396-f006:**
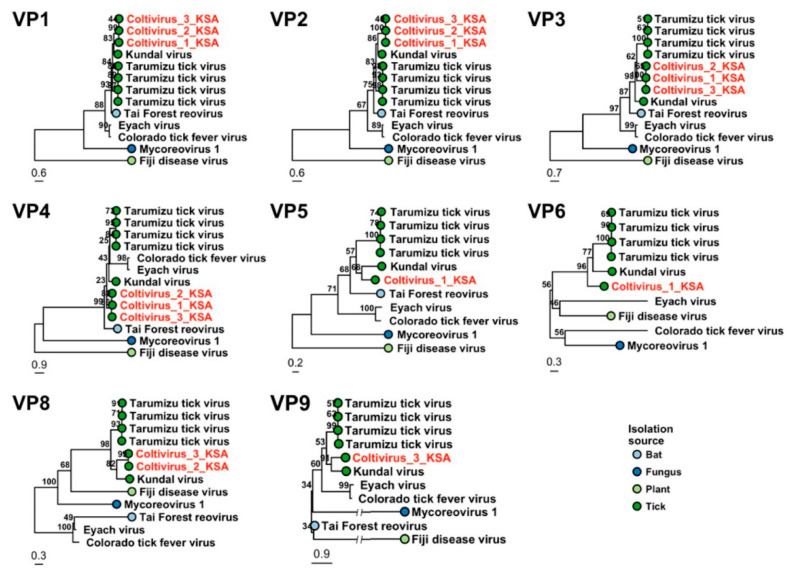
Phylogenetic trees based on VP1 to VP6, VP8 and VP9 genes of genus *Coltivirus* The viruses detected in Saudi Arabia are marked with red color.

**Table 1 viruses-13-01396-t001:** Summary of virus families, genera, species number of reads, percentage of identity (amino acids) and number of contigs detected.

Family	Genus	Species (Closest Match)	Read (*n*)	Amino Acid P-Identity (%)	Contig (*n*)
Phenuiviridae	Phlebovirus	Iftin virus	64,673	90.68–100	25
	Bandavirus	Guertu bandavirus	1918	87–95	7
Nairoviridae	Orthonairovirus	CCHFV	2917	96.88–99.67	15
	Orthonairovirus	Tamdy virus	7144	97.7–100	17
Reoviridae	Coltivirus	Kundal coltivirus	9340	43.57–71.37	11
	Coltivirus	Tarumizu coltivirus	2138	39.68–65.07	7
Chuviridae	Mivirus	Bole mivirus	918	74.8	6
Totiviridae	Totivirus	Lonestar tick totivirus	1156	42.65 and 55.56%	5
Flaviviridae	Pestivirus	Bole tick Virus 4	820	81.10%	10
unknown	unknown	Hubei tick virus 1	31,363	73.45	29
unknown	unknown	Gerbovich virus	19,409	74.3–76	23
unknown	unknown	Bole tick virus 1	820	83–84	10

## Data Availability

The sequences of viruses were deposited at Genbank under the accession codes (MZ566475 to MZ566491 and MZ567075 to MZ567079).

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
