# Peer review of "Viral RNA Metagenomics of Hyalomma Ticks Collected from Dromedary Camels in Makkah Province, Saudi Arabia"

_viruses, 2021, doi:10.3390/v13071396_

Round 1

Reviewer 1 Report

Title: suggest to have a appropriate title and include specific word 'metagenomic study' and what type of  viral composition authors have analyzed.

Under Research Methodology line 126: not mentioned how the tick pools are being named and is not supported this data by a figure or a table.

Under Result: line 163: suggest to present only the result. 

line325: repetition of year

Table 1: title and the information given does not match 

Discussion: suggest to mention the absence of important virus like MERS CoV, alpha corona virus etc. what about DNA viruses?

References: requires uniform formatting.

Author Response

Comments and suggestions of Reviewer 1:

Comment 1:

Title: suggest to have a appropriate title and include specific word 'metagenomic study' and what type of viral composition authors have analyzed.

Response to comment 1:

We thank the referee for his comment and the title has been changed to “Viral RNA metagenomics of Hyalomma ticks collected from dromedary camels in Makkah Province, Saudi Arabia”. We did not want to make the title longer than 20 words count.

Comment 2:

Under Research Methodology line 126: not mentioned how the tick pools are being named and is not supported this data by a figure or a table.

Response to comment 2:

We are sorry for this inconvenience and details have been added to the mentioned part. After ticks’ barcoding, we found that all the ticks are belonging to Hyalomma sp, so pooling was done according to the geographic region: Al-Khurma (pools K1 to K10) and Al-Taif (pools T1 to T10). Engorged ticks were pooled separately. The first 3 pools of each collection site contained engorged ticks and the rest of pools contained non-engorged ticks.

Comment 3:

Under Result:

line 163: suggest to present only the result. 

line325: repetition of year

Response to comment 3:

Line 163: Revised as suggested.

Line 325: repetition of the year is removed.

Comment 4:

Table 1: title and the information given does not match 

Response to comment 4:

The title has been revised. In addition, we revised the figure legends.

Comment 5:

Discussion: suggest to mention the absence of important virus like MERS CoV, alpha corona virus etc. what about DNA viruses?

Response to comment 5:

The absence of coronaviruses is now mentioned in Discussion section (line 287).

Regarding DNA viruses, we used RNA as a starting material for the library preparation. While using this approach, it may be possible to detect also DNA viruses with active transcription, latent DNA virus infections will be missed. Therefore, we have now indicated more clearly that we are discussing RNA viromes throughout the manuscript. 

Comment 6:

References: requires uniform formatting.

Response to comment 6:

This has been considered in the revised version

Reviewer 2 Report

This is a well presented and interesting manuscript identifying diverse viruses harbored by Hyalomma ticks sampled from camels in Saudi Arabia. The findings are relevant not only to the study region, but tick-borne disease and emerging virus researchers globally.

I recommend its publication after minor, mainly language, revisions. Below are specific comment:

Line 3: “Province” should be capitalized

Line 25: ‘human’ should be ‘humans’

Line 29: “We deciphered the virome.” This phrase leaves the reader wondering how. No details on the methods employed are presented in the abstract. The abstract requires a clear statement on the metagenomic approach employed to characterize the virome.

Line 42: The ‘the’ before ‘human’ is unnecessary.

Line 50: Change to “mortality rates in humans’

Line 52-53: “dengue” and “chikungunya” should not be capitalized as they are not named after geographical names; ‘virus’ and ‘hemorrhagic’ should not be capitalized either

Line 59: Boophilus is an old term for some Rhipicephalus species and is thus redundant here.

Line 59: Change to “…ticks infecting small ruminants….”

Line 73: Delete ‘on’ before’camels’

Line 76: ‘human’ should be ‘humans’

Line 77: Delete ‘all’ …. Also, why are there so many strange changing fonts in this section?

Line 84-85: Combine paragraphs

Line 87: Change to “Such approaches can enable timely responses to…”

Line 95: “huge number”..This is colloquial and not clear, why not provide a citable estimate?

Line 110: Change to ‘604-bp fragment’

Line 124: Change to ‘5-mm’. Such things need to be hyphenated as they describe the following noun (stainless steel beads)

Line 131: “Phlebovirus’ needs to be capitalized and italicized.

Line 186: Capitalize “District”

Lines 274-275: Change to “…identified as H. dromedarii and H. impeltatum, as in previous tick prevalence studies…”

Line 276: ‘here’ is unnecessary

Line 294: Change to “febrile illness in humans”

Lines 295-296: Delete ‘associated’

Lines 298-299: Not clear what is meant by “genogroup associated to TOV” Do you mean phylogenetically close to?

Lines 310-311:  Change to “…people in Guertu County, China, have…”

Line 317: ‘coltivirus’ should not be capitalized

Line 335: Do not abbreviate genus names if in the beginning of a sentence. “Hyalomma” should be written out.

Line 335: I believe with ‘displayed’ you mean ‘harbored’

Line 340: ‘which’ should be ‘that’

Author Response

Comments and suggestions of Reviewer 2:

Comment 1:

This is a well presented and interesting manuscript identifying diverse viruses harbored by Hyalomma ticks sampled from camels in Saudi Arabia. The findings are relevant not only to the study region, but tick-borne disease and emerging virus researchers globally.

Response to comment 1:

We thank the referee for this comment and we appreciate his suggestions.

Comment 2:

I recommend its publication after minor, mainly language, revisions. Below are specific comment:

Response to comment 2:

All the requested modifications are considered.

Line 3: “Province” should be capitalized

 This is done.

Line 25: ‘human’ should be ‘humans’

 This is done.

Line 29: “We deciphered the virome.” This phrase leaves the reader wondering how. No details on the methods employed are presented in the abstract. The abstract requires a clear statement on the metagenomic approach employed to characterize the virome.

We thank the referee for this comment and we mentioned the requested modification in the revised manuscript.

Line 42: The ‘the’ before ‘human’ is unnecessary.

“ The” has been removed.

Line 50: Change to “mortality rates in humans’

This has been done 

Line 52-53: “dengue” and “chikungunya” should not be capitalized as they are not named after geographical names; ‘virus’ and ‘hemorrhagic’ should not be capitalized either

Revised as suggested

Line 59: Boophilus is an old term for some Rhipicephalus species and is thus redundant here.

 We thank the referee for this comment and Boophilus has been removed to avoid redundancy in the text.

Line 59: Change to “…ticks infecting small ruminants….”

 We have changed it to “ticks infesting small ruminants…”

Line 73: Delete ‘on’ before’camels’

 As requested, ‘on’ is deleted.

Line 76: ‘human’ should be ‘humans’

This is done. 

Line 77: Delete ‘all’ …. Also, why are there so many strange changing fonts in this section?

 ‘all’ is deleted and the font has been adjusted.

Line 84-85: Combine paragraphs

 This is done. 

Line 87: Change to “Such approaches can enable timely responses to…”

 This has been modified.

Line 95: “huge number”..This is colloquial and not clear, why not provide a citable estimate?

 This has been changed to “more than 1,2 million sheep, goats and camel annually during pilgrimage” and a new reference is added.

Line 110: Change to ‘604-bp fragment’

 This is done. 

Line 124: Change to ‘5-mm’. Such things need to be hyphenated as they describe the following noun (stainless steel beads)

 This is done.

Line 131: “Phlebovirus’ needs to be capitalized and italicized.

 This is done.

Line 186: Capitalize “District”

 The word “District” is capitalized.

Lines 274-275: Change to “…identified as H. dromedarii and H. impeltatum, as in previous tick prevalence studies…”

 This is done.

Line 276: ‘here’ is unnecessary

 ‘here’ is removed

Line 294: Change to “febrile illness in humans”

  This is done.

Lines 295-296: Delete ‘associated’

‘associated’ is deleted. 

Lines 298-299: Not clear what is meant by “genogroup associated to TOV” Do you mean phylogenetically close to?

Yes, we mean phylogenetically related. The last two sentences of this paragraph were modified as follows:

“In addition, two virus species phylogenetically related to TOV, Tacheng tick virus 1 (TcTV1) and Songling virus have caused febrile illness in humans [23, 40]. Most recently, a novel tick-associated virus related to TOV, Sulina virus, has been detected in Ixodes ricinus ticks in the Danube River Delta, in Europe [41].”

Lines 310-311:  Change to “…people in Guertu County, China, have…”

 This modification is done.

Line 317: ‘coltivirus’ should not be capitalized

 This is done.

Line 335: Do not abbreviate genus names if in the beginning of a sentence. “Hyalomma” should be written out.

This is done.

Line 335: I believe with ‘displayed’ you mean ‘harbored’

 ‘displayed’ is replaced by ‘harbored’.

Line 340: ‘which’ should be ‘that’

‘which’ is replaced by ‘that’.